# The Mortality Attributable to Candidemia in *C. auris* Is Higher than That in Other *Candida* Species: Myth or Reality?

**DOI:** 10.3390/jof9040430

**Published:** 2023-03-31

**Authors:** Carlos A. Alvarez-Moreno, Soraya Morales-López, Gerson J. Rodriguez, Jose Y. Rodriguez, Estelle Robert, Carine Picot, Andrés Ceballos-Garzon, Claudia M. Parra-Giraldo, Patrice Le Pape

**Affiliations:** 1Facultad de Medicina, Universidad Nacional de Colombia, Clínica Universitaria Colombia, Clínica Colsanitas, Bogotá 111321, Colombia; 2Grupo CINBIOS, Programa de Microbiología, Universidad Popular del Cesar, Valledupar 200004, Colombia; 3Centro de Investigaciones Microbiológicas del Cesar (CIMCE), Valledupar 200002, Colombia; 4Cibles et Médicaments des Infections et de l’Immunité, Nantes Université, CHU de Nantes, IICiMed, 10 UR1155, 44000 Nantes, France; 5Unidad de Investigacion en Proteómica y Micosis Humanas, Grupo de investigacion en Enfermedades Infecciosas, Departamento de Microbiología, Facultad de Ciencias Pontificia Universidad Javeriana, Bogotá 110231, Colombia

**Keywords:** *Candida auris*, mortality, candidemia, Colombia

## Abstract

*Candida auris* has become a major health threat due to its transmissibility, multidrug resistance and severe outcomes. In a case-control design, 74 hospitalised patients with candidemia were enrolled. In total, 22 cases (29.7%) and 52 controls (*C. albicans*, 21.6%; *C. parapsilosis*, 21.6%; *C. tropicalis*, 21.6%; *C. glabrata*, 1.4%) were included and analysed in this study. Risk factors, clinical and microbiological characteristics and outcomes of patients with *C. auris* and non-*auris Candida* species (NACS) candidemia were compared. Previous fluconazole exposure was significantly higher in *C. auris* candidemia patients (OR 3.3; 1.15–9.5). Most *C. auris* isolates were resistant to fluconazole (86.3%) and amphotericin B (59%) whilst NACS isolates were generally susceptible. No isolates resistant to echinocandins were detected. The average time to start antifungal therapy was 3.6 days. Sixty-three (85.1%) patients received adequate antifungal therapy, without significant differences between the two groups. The crude mortality at 30 and 90 days of candidemia was up to 37.8% and 40.5%, respectively. However, there was no difference in mortality both at 30 and 90 days between the group with candidemia by *C. auris* (31.8%) and by NACS (42.3%) (OR 0.6; 95% IC 0.24–1.97) and 36.4% and 42.3% (0.77; 0.27–2.1), respectively. In this study, mortality due to candidemia between *C. auris* and NACS was similar. Appropriate antifungal therapy in both groups may have contributed to finding no differences in outcomes.

## 1. Introduction

Candidemia remains a worldwide public health concern. In Latin America, the incidence varies between 0.74–6.0 per 1000 hospital admissions and is associated with a high mortality rate (30–76%) [1]. Although there are geographical differences in the distribution of *Candida* species, *Candida albicans* remains the most frequently isolated. However, there is an increase in the incidence of invasive infections by non-*albicans Candida* species, including *Candida auris* [1,2]. *C. auris* is recognised as an emerging pathogen, which is difficult to identify, multiple-drug-resistant and highly transmissible. *C. auris* has a high potential for outbreaks in healthcare settings, possibly due to environmental contamination or transient colonisation by people or medical devices [3,4]. This species is known to form biofilms on inert surfaces, which can persist for extended periods and require stringent disinfection processes to remove [5,6]. To adapt to dry abiotic environments, *C. auris* activates stress-activated proteins [7] and produces hydrolytic enzymes that protect the yeast from environmental stressors and disinfectants such as chlorhexidine and hydrogen peroxide. Chlorine-based products have been shown to be the most effective for environmental surface disinfection [8,9,10].

*C. auris* was described as a new species in 2009, when isolated from the external ear of a woman in Japan [11]. In 2011, it was described for the first time as a cause of fungemia in South Korea [12]. It has subsequently been isolated in at least 39 countries on 5 continents, causing isolated cases of colonisation to true outbreaks of invasive infections such as candidemia [13]. The first outbreak of *C. auris* in the Americas was reported in Venezuela (2012–2013) [14]. In Colombia, isolated cases have been reported since 2012 and, through a retrospective analysis, outbreaks since 2016 have indicated that *C. auris* could be considered to be endemic in several cities since 2013 [15,16,17]. In October 2022, the WHO issued its first fungal priority pathogen list, which included *C. auris* among others [18]. *C. auris* is commonly resistant to azole drugs and isolates that are resistant to all three main classes of antifungal agents have also been reported [19]. Mutations in the drug-target lanosterol 14-α-demethylase ERG11 gene such as Y132F, K143R and F126L frequently lead to azole resistance in *C. auris* strains. Additionally, mutations in the transcription factor TAC1 can cause an overexpression of CDR1, which also leads to high-level azole resistance in *C. auris* [20].

Mortality due to fungemia by *C. auris* is high. Recently, Chen et al. conducted a meta-analysis and found that the overall mortality of *C. auris* infections was 39% and 45% for bloodstream infections (BSI) [21]. However, the same authors conclude that the observed heterogeneities such as clade, BSI, drug resistance, continent and publication year were limitations. In general, mortality could be higher than that produced by other Candida species as a result of characteristics such as multiresistance, specific virulence factors, phenotypes, the ability to produce biofilms and a greater association with candidemia [13,17,21,22,23]. In addition, it has been difficult to confirm whether mortality from *C. auris* is related to the early selection of an adequate antifungal treatment due to its multidrug resistance. The objective of this study was to assess mortality due to *C. auris* fungemia and compare it with that resulting from NACS fungemia in the same period of time and in the same hospitals.

## 2. Materials and Methods

A case–case study was conducted in 12 hospitals in Valledupar, Colombia. We included all patients diagnosed with candidemia, defined by the presence of at least one positive blood culture for *Candida* spp. The participating institutions carried out laboratory surveillance for candidemia. All hospitals had automated blood culture systems. Blood cultures were collected under aseptic conditions and processed by conventional automated systems (BacT/ALERT 3D, bioMerieux, Marcy l’Etoile, France). The initial identification was performed with the method available at each institution of origin; i.e., Vitek 2 (bioMérieux, Marcy l’Etoile, France), Phoenix (Becton Dickinson, Franklin Lakes, NJ, USA), Austoscan-4 (Beckman Coulter, Fullerton, CA, USA) and Api Candida (BioMérieux, Marcy l’Etoile, France). The isolates were then sent to a regional reference laboratory where they were seeded in CHROMagar Candida medium (CHROMagar, Paris, France) and identified using MALDI-TOF mass spectrometry (Bruker Daltonics, Billerica, MA, USA). MALDI-TOF MS results were obtained according to the manufacturer’s specifications and all clinical isolates had a score above 2.0 [24]. *C. auris* identifications were confirmed by a single-tube PCR method based on the amplification of internal transcribed spacer (ITS) regions, as previously described by us [25]. Susceptibility to widely used antifungal drugs (AFG, anidulafungin; MCF, micafungin; CAS, caspofungin; 5-FC, flucytosine; PSC, posaconazole; VRC, voriconazole; ISV, isavuconazole; FLC, fluconazole; AMB, amphotericin B) was determined by Sensititre Yeast One^®^ AST plates (ThermoFisher Scientific, Les Ulis, France). This method was used because it remains comparable with the CLSI reference method for testing the susceptibility of *Candida* spp. [26]. For the NACS isolates (*C. albicans*, *C. tropicalis*, *C. glabrata* and *C. parapsilosis*), CLSI M27-A3 breakpoints were used [27]. In the case of *C. auris*, breakpoints recommended by the US Centers for Disease Control and Prevention (CDC) were applied. Resistance to FLC was set at ≥32 µg/mL, AMB at ≥2.0 µg/mL and CAS at ≥2 µg/mL [28].

The clinical histories of patients with candidemia were reviewed; the information was collected in a case report and then tabulated into a database designed for this study. The demographic, clinical and microbiological variables as well as antifungal prophylaxis (FLC) and antifungal treatments were included. The risk factors for the development of fungemia that were evaluated were diabetes mellitus, haematological malignancy, solid organ malignancy, HIV infection, renal failure, need for haemodialysis, solid organ transplantation, extensive burns, need for mechanical ventilation, central venous catheter (CVC) or bladder catheter, red blood cell transfusion, history of abdominal surgery 30 days before the development of fungemia, parenteral nutrition, use of steroids defined as more than 20 mg of prednisone per day or an equivalent corticosteroid for more than 14 days before the onset of fungemia and previous use of antibiotics for more than 48 h in the 30 days before the onset of fungemia. The time to development of fungemia was defined as the number of days from hospitalisation to positive blood cultures for yeast. The time to start antifungal therapy was defined as the days from the first positive blood culture for yeast to the start of adequate antifungal therapy. An appropriate antifungal therapy was defined according to antimicrobial susceptibility tests. Additionally, information on outcomes such as mortality at 30 and 60 days from the onset of fungemia was included.

The data for categorical variables were expressed as a percentage and the continuous variables were expressed as the mean ± SD or the median (interquartile range). A chi-squared test or Fisher’s exact test (two-tailed) was used to compare the categorical variables and an unpaired Student’s *t*-test was used to compare the continuous variables. A Kaplan–Meier analysis and log-rank test, recorded as a hazard ratio (HR: 95% CI), were used to compare both 30- and 90-day survival between the *C. auris* and NACS groups. Multivariate, backward, stepwise and logistic regression analyses were used to identify independent risk factors associated with day 30 mortality of patients; the results were presented as odds ratios (ORs) with 95% confidence intervals (95% CIs) and *p*-values < 0.05 were considered to be significant. The statistical analysis was performed using EPI Info 7™ and GraphPad Prism 8.4.3. (Dotmatics, Boston, MA, USA) software.

## 3. Results

Seventy-four patients with candidemia were enrolled. The mean age of the patients was 38.2 years (range 5 days–88 years; SD 28.7); 44 patients were male. The average number of days from the start of hospitalisation until the onset of fungemia was 18.5 (range 0–55; SD 13.4). A total of 22 *C. auris* cases (29.7%) and 52 NACS cases (*C. albicans*, 25.6%; *C. parapsilosis*, 21.6%; *C. tropicalis*, 21.6%; *C. glabrata*, 1.4%) were included and analysed in this study.

The comparison of the main clinical and epidemiological variables is summarised in Table 1. Previous FLC exposure was significantly higher in *C. auris* candidemia patients (OR 3.3; 1.15–9.5). The in vitro antifungal susceptibility of the *Candida* species isolates is presented in Table 2. Most *C. auris* isolates were resistant to FLC (86.3%) and AMB (59%) whereas all were susceptible to echinocandins. Echinocandins (MCF and CAS) and 5-FC showed the lowest MICs among the antifungal drugs tested, with 100% of tested *C. auris* isolates having MICs of ≤0.125 µg/mL. Regarding the NACS isolates, these were generally susceptible (FLC-resistant, 9.6%; AMB-resistant, 3.8%). No echinocandin-resistant isolates were detected.

A total of 9 (12.2%) patients did not receive antifungal treatment, 2 in the *C. auris* group and 7 in the NACS group. Among the 65 (87.8%) patients who received antifungal treatments, 48.6% received CAS, 29.7% FLC, 5.4% AMB and 4.1% a combined antifungal therapy. The use of CAS as a fungemia treatment was higher in patients with *C. auris* (86.4% vs. 32.7%). The average time to start antifungal therapy was 3.6 days. A total of 63 (85.1%) patients received adequate antifungal therapy, without significant differences between the 2 groups (*C. auris*, 86.4% vs. NACS, 84.6%) (*p*-value 1.1; OR 0.27–4.8). The crude mortality at 30 and 90 days of candidemia was up to 37.8% and 40.5%, respectively. However, there was no difference in mortality at either 30 or 90 days between the group with candidemia caused by *C. auris* and the group with candidemia caused by NCAS (31.8% vs. 40.4%; OR 0.6; 95% CI 0.24–1.97 and 36.4% vs. 42.3%; 0.77; 0.27–2.1, respectively.

## 4. Discussion

An epidemiologic study conducted between 2008 and 2010 on candidemia in Latin America found that in Colombia, *C. parapsilosis* (38.5%) and *C. albicans* (36.7%) predominated, followed by *C. tropicalis* (17.4%), *C. glabrata* (4.6%) and *C. guilliermondii* (1.8%) [1]. However, in the last years, the situation has changed with the emergence of *C. auris* throughout the country [20,29]. Despite cases of infections by *C. auris* being described in several cities, most of the cases have been concentrated on the Atlantic coast [16,29].

This prospective study was performed during 2016–2017 at 12 health institutions in Valledupar, Cesar, a city of 450,000 inhabitants located in northern Colombia (approximately 180 km from the Atlantic coast). Seventy-four patients with candidemia were enrolled. *C. auris* was responsible for 29.7% of the fungemia, leading *C. auris* to be the most prevalent species in the area. The demographic and clinical characteristics of patients with *Candida* spp fungemia were similar to those reported in other studies [1,14]. There was no difference in risk factors for the development of fungemia in the group with *C. auris* vs. NACS, except for the previous use of FLC, which was higher in the *C. auris* group. This could be explained by the selection pressure exerted by this azole drug. In line with previous studies, the risk factors were not different from those associated with invasive infection due to NACS [30].

In the present study, 12.2% of patients did not receive antifungal treatment. The most likely reason was a late diagnosis as most of these patients died very early after the diagnosis of fungemia. CAS was the main antifungal given as the primary treatment. The *C. auris* group received CAS more frequently than the NACS fungemia group, probably as a consequence of the FLC prophylaxis that was more frequent in this group.

The literature has reported that mortality rates due to invasive *C. auris* infections range from 30% to 72% [31,32,33,34]. However, as in our study, in the multicentre retrospective case-control study of Simon et al., *C. auris* bloodstream infections were not associated with increased 30 day or 90 day mortality [35]. Indeed, in our study, mortality due to candidemia was similar between *C. auris* and NACS. An appropriate antifungal therapy in both groups may have contributed to finding no differences in outcomes. Similarly, comparative animal models, including murine and invertebrate models, indicated that *C. auris* was less virulent than *C. albicans*. Based on these models [36], it has been suggested that the decrease in virulence is likely related to the inability of *C. auris* to develop hyphae or pseudohyphae in mammals, which play a fundamental role in tissue invasion [37].

Although this study was conducted before the COVID-19 outbreak, we previously reported 20 cases of fungemia in hospitalised patients with severe acute respiratory syndrome coronavirus 2 (SARS-CoV-2) infection across 4 institutions in the northern region of Colombia between June and September 2020 [38]. Of these, six patients had fungemia caused by *C. auris*. In this case series, the time to develop fungemia was similar to that reported in the current article (17.7 vs. 18.4 days). However, the mortality rate for this group of patients was higher (60% vs. 37.8%). Similarly, Villanueva-Lozano found a mortality rate of 83.3% (5/6) among patients with fungemia during an outbreak of *C. auris* infection in a COVID-19 hospital in Mexico [39] and Chowdhary found a mortality rate of 60% in patients with candidemia caused by *C. auris* who were critically ill with coronavirus disease and admitted to an intensive care unit between April and July 2020 in New Delhi, India [40]. Therefore, it is expected that patients with fungemia caused by *C. auris* and SARS-CoV-2 infection may have a higher mortality rate.

Although *C. auris* has been described as a multidrug-resistant pathogen capable of generating resistance to the three most important classes of available antifungal drugs, there are geographical differences in the pattern of resistance [29]. A study in the USA found that more than 90% of *C. auris* isolates were resistant to FLC, more than 60% were resistant to AMB, 3.9% were resistant to echinocandins and 3 isolates were found to be pan-resistant [41]. In contrast, a report from India found that 40% of the isolates displayed a high MIC to CAS [42]. Our strains were characterised by a high percentage of resistance to azoles and AMB, placing echinocandins as the only available option for the treatment of these infections at the moment.

However, the emergence of echinocandin resistance in *C. auris* has become a major concern in several countries and the larger use of this drug to control nosocomial outbreaks could complicate patient management in the future [43]. This risk of multi-resistance explains why it is important to know the resistance levels of circulating clinical isolates and, therefore, to have validated susceptibility assays [44].

Although the mortality of *C. auris* candidemia was not higher than for other *Candida* species, this species can cause intra-hospital outbreaks. This is why limiting the transmission of this microorganism, adherence to infection control programs (contact isolation and hand washing) and adequate cleaning and environmental disinfection protocols should be emphasised.

## Figures and Tables

**Table 1 jof-09-00430-t001:** Epidemiological and demographic characteristics, underlying conditions, treatments and outcomes of patients with fungemia in 12 hospitals in Valledupar, Colombia.

	Total	%/(Range) SD	*C. auris*	%/(Range) SD	No. *auris*	%/(Range) SD	*p*-Value (OR)
*n*	74	100%	22	29.7	52	70.3	
Age (years), median	38.2	(5 days to 88 years old) SD 28.7	43.3	(8 months to 77 years old) SD 21.1	36	(5 days to 88 years old) SD 31.3	0.32 * (−7.3–21.8) *
Gender, male	44	59.5	12	54.5	32	61.5	0.3 (0.66–2.09)
Diabetes mellitus	9	12.2	4	18.2	5	9.6	0.6 (0.7–3.04)
Haematological malignancy	2	2.7	0	0.0	2	3.8	NA
Renal failure	21	28.4	9	40.9	12	23.1	0.1 (0.8–3.5)
Dialysis	11	14.9	6	27.3	5	9.6	0.06 (0.8–3.8)
Transplant	1	1.4	0	0.0	1	1.9	NA
Pancreatitis	3	4.1	2	9.1	1	1.9	0.08 (0.9–5.4)
Solid tumour	8	10.8	3	13.6	5	9.6	0.16 (0.8–44)
Extensive burns	1	1.4	0	0.0	1	1.9	NA
Mechanical ventilation	46	62.2	16	72.7	30	57.7	0.16 (0.76–3.1)
Blood transfusion	39	52.7	13	59.1	26	50.0	0.3 (0.68–2.8)
HIV	5	6.8	3	13.6	2	3.8	0.04 (0.99–5.32)
Central venous catheter	60	81.1	20	90.9	40	76.9	0.13 (0.74–3.3)
Urinary catheter	51	68.9	18	81.8	33	63.5	0.09 (0.81–3.4)
Abdominal surgery	24	32.4	7	31.8	17	32.7	0.09 (0.82–3.9)
Parenteral nutrition	29	39.2	6	27.3	23	44.2	0.48
Receipt of corticosteroids	8	10.8	2	9.1	6	11.5	0.13 (0.26–8.7)
Receipt of antibiotics	68	91.9	19	86.4	49	94.2	0.3 (0.07–2.9)
Previous use of fluconazole	23	31.1	11	50.0	19	36.5	0.02 (3.3–9.5)
Duration (days) of hospitalisation before candidemia, median (range)	18.4	(0−55) SD 13.4	21	(0–55) SD 17.4	17.3	(0–48) SD 11.3	0.29 (−3.1–10.3) *
Treatment							
No treatment	9	12.2	2	9.1	7	13.5	
Fluconazole	22	29.7	1	4.5	21	40.4	
Caspofungin	36	48.6	19	86.4	17	32.7	
Amphotericin B	4	5.4	0	0.0	4	7.7	
Combination of antifungals	3	4.1	0	0.0	3	5.8	
Time to start antifungal (days)	3.6	(−1–34) SD 4.6	3.2	(1–6) SD 8.0	3.8	(−1–34) SD 5.4	0.68 (−2.9–1.98) *
Appropriate antifungal treatment	63	85.1	19	86.4	44	84.6	1.1 (0.27–4.8)
30 day mortality	28	37.8	7	31.8	21	40.4	0.66 (0.24–1.97)
90 day mortality	30	40.5	8	36.4	22	42.3	0.77 (0.27–2.1)

* Two-tailed *p*-value and 95% confidence interval.

**Table 2 jof-09-00430-t002:** Antifungal activities of the nine tested drugs against *C. auris* (*n* = 22) and NACS (*n* = 52).

Species	Drug	Number (and Cumulative Percentage) of *Candida* spp. Strains with MIC (µg/mL)																																					
		≤0.015		0.03		0.06			0.125			0.25			0.50			1			2			4			8			16		32		64			≥128		
** *C. auris* ** ***n* = 22 **	ANF	6	(27)	**3**	**(41)**		6	(68)		6	(95)		1	(100)																									
	MCF	6	(27)	**5**	**(50)**		4	(68)		7	(100)																												
	CAS	7	(32)	**6**	**(59)**		6	(86)		3	(100)																												
	5-FC						**19**	**(86)**		3	(100)																												
	PSC	3	(14)	4	(32)		1	(36)		** 5 **	** (59) **		8	(95)		1	(100)																						
	VRC	1	(5)	2	(14)		3	(28)		2	(37)		**4**	**(55)**		7	(87)		3	(100)																			
	ISV			1	(5)		4	(22)		2	(31)		**8**	**(67)**		7	(100)																						
	FLC																								2	(9)					1	(14)	5	(38)	**5**	**(55)**		9	(100)
	AMB															4	(18)		5	(41)		** 12 **	** (96) **		1	(100)													
*C. albicans* *n* = 19	ANF	**16**	**(84)**	2	(95)											1	(100)																						
	MCF	** 18 **	** (95) **													1	(100)																						
	CAS	5	(26)	**11**	**(79)**		2	(95)								1	(100)																						
	5-FC	**16**	**(84)**							1	(89)					1	(95)		1	(100)																			
	PSC	**13**	**(68)**	5	(95)														1	(100)																			
	VRC	** 18 **	** (95) **										1	(100)																									
	ISV	6	(32)	**9**	**(79)**		3	(95)								1	(100)																						
	FLC									2	(11)		**9**	**(58)**		6	(89)		1	(95)								1	(100)										
	AMB															**11**	**(58)**		8	(100)																			
*C. tropicalis* *n* = 16	ANF	4	(25)	2	(13)		**2**	**(50)**		7	(94)		1	(100)																									
	MCF	6	(38)	** 10 **	** (100) **																																		
	CAS	1	(6)	**13**	**(88)**		2	(100)																															
	5-FC	**13**	**(81)**							1	(87)																								2	(100)			
	PSC									**9**	**(56)**		6	(94)		1	(100)																						
	VRC			3	(19)					**10**	**(81)**		2	(75)		1	(100)																						
	ISV									**8**	**(50)**		7	(94)		1	(100)																						
	FLC															1	(6)		**9**	**(63)**		5	(94)					1	(100)										
	AMB															1	(6)		**13**	**(88)**		1	(94)		1	(100)													
*C. parapsilosis* *n* = 16	ANF						1	(6)		1	(13)					3	(31)		**8**	**(75)**		3	(100)																
	MCF						1	(6)					1	(13)		4	(38)		**7**	**(81)**		3	(100)																
	CAS						2	(13)		1	(19)		**8**	**(69)**		3	(88)		2	(100)																			
	5-FC						** 15 **	** (94) **											1	(100)																			
	PSC	4	(25)	**6**	**(63)**		4	(88)		1	(94)					1	(100)																						
	VRC	7	(44)	**1**	**(50)**		4	(75)		1	(81)		1	(88)		1	(94)		1	(100)																			
	ISV			6	(38)		**7**	**(81)**		2	(94)		1	(100)																									
	FLC															4	(25)		3	(44)		**4**	**(69)**		2	(81)		2	(94)				1	(100)					
	AMB												2	(13)		**12**	**(88)**		2	(100)																			
*C. glabrata* *n* = 1	ANF	1	(100)																																				
	MCF	1	(100)																																				
	CAS						1	(100)																															
	5-FC						1	(100)																															
	PSC																		1	(100)																			
	VRC															1	(100)																						
	ISV															1	(100)																						
	FLC																														1	(100)							
	AMB																		1	(100)																			

ANF: anidulafungin; MCF: micafungin; CAS: caspofungin; 5-FC: 5-fluorocytosine; PSC: posaconazole; VRC: voriconazole; ISV: isavuconazole; FLC: fluconazole; AMB: amphotericin B. MIC_50_s (minimal inhibitory concentration at which ≥50% of the strains are inhibited) and MIC_90_s (minimal inhibitory concentration at which ≥90% of the strains are inhibited) are depicted in bold letters and are underlined, respectively.

## Data Availability

Not applicable.

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
