# Peer review of "The Mortality Attributable to Candidemia in C. auris Is Higher than That in Other Candida Species: Myth or Reality?"

_jof, 2023, doi:10.3390/jof9040430_

Round 1
Reviewer 1 Report
Very well written and succinct.
Author Response
Reviewer #1:
Very well written and succinct.

Reviewer 2 Report
General comments
It is a well written MS. The Authors confirmed that overall mortality of candidemia was similar between C. auris versus other Candida species in 12 hospitals in Valledupar, Colombia. The C. auris isolates probably belonged to the S. American clade (it can be mention in the MS).
Minor comments
1. The retrospective study performed before the COVID-19 outbreak. However, during the COVID outbreak other Authors (Villanueva-Lozano et al, doi: 10.1016/j.cmi.2020.12.030., Chowdhary et al, DOI: 10.3201/eid2611.203504 and Hanson et al, DOI: 10.1128/AAC.01146-21) with 3 different clades, including the S. American clade have found a significantly higher mortality rate. These facts can be mention in the Discussion.
2. In animal model the virulence of different C. auris clades was significantly lower than C. albicans (doi: 10.1080/22221751.2020.1771218). As the number of C. albicans candidemia (25.6%) was comparable to C. auris candidemia (29.7%), it may be worth to calculate and discuss the mortality (at least numerically) due to C. albicans.
Author Response
Minor comments
- The retrospective study performed before the COVID-19 outbreak. However, during the COVID outbreak other Authors (Villanueva-Lozano et al, doi: 10.1016/j.cmi.2020.12.030., Chowdhary et al, DOI: 10.3201/eid2611.203504 and Hanson et al, DOI: 1128/AAC.01146-21) with 3 different clades, including the S. American clade have found a significantly higher mortality rate. These facts can be mention in the Discussion.
Answer:
The paragraph was added:
“Although this study was conducted before the COVID-19 outbreak, we previously reported 20 cases of fungemia in hospitalized patients with severe acute respiratory syndrome coronavirus 2 (SARS-CoV-2) infection across four institutions in the northern region of Colombia between June and September 2020 (35). Of these, six patients had fungemia caused by C. auris. In this case series, the time to develop fungemia was similar to that reported in the current article (17.7 vs. 18.4 days). However, the mortality rate for this group of patients was higher (60% vs. 37.8%). Similarly, Villanueva-Lozano found a mortality rate of 83.3% (5/6) among patients with fungemia during an outbreak of C. auris infection in a COVID-19 hospital in Mexico (36), and Chowdhary found a mortality rate of 60% in patients with candidemia caused by C. auris who were critically ill with coronavirus disease and admitted to an intensive care unit between April and July 2020 in New Delhi, India (37). Therefore, it is expected that patients with fungemia caused by C. auris and SARS-CoV-2 infection may have a higher mortality rate.”
- In animal model the virulence of different C. auris clades was significantly lower than C. albicans (doi: 1080/22221751.2020.1771218). As the number of C. albicans candidemia (25.6%) was comparable to C. auris candidemia (29.7%), it may be worth to calculate and discuss the mortality (at least numerically) due to C. albicans.
Answer:
The paragraph was added:
“Similarly, comparative animal models, including murine and invertebrate models, indi-cate that C. auris is less virulent than C. albicans. Based on these models (35), it has been suggested that the decrease in virulence is likely related to C. auris' inability to develop hyphae or pseudohyphae in mammals, which play a fundamental role in tissue invasion

Reviewer 3 Report
The study by Carlos Alvarez-Moreno et al. about mortality caused by Candida auris compared to other candida species is well presented in this study and referred to a defined area of Colombia. According to their data, there is no difference in mortality although previous studies supported different results. This is interesting for people who take care, to have always in mind different risk factors and choose each time appropriate treatment.
My suggestion is to be accepted after minor revision concerning candida species identification. These data have to be presented either in the main text or as a supplementary file, to be clear that identification was done correctly.
Author Response
My suggestion is to be accepted after minor revision concerning candida species identification. These data have to be presented either in the main text or as a supplementary file, to be clear that identification was done correctly.
Answer:
The paragraph was added:
“Then, the isolates were sent to a regional reference laboratory where they were seeded in CHROMagar Candida Medium (CHROMagar, Paris, France) and identified using MALDI-TOF mass spectrometry (Bruker Daltonics,Billerica, USA). MALDI-TOF MS results were obtained according manufacturer’s specifications and all clinical isolates had a score above 2.0 (24). C. auris identifications were confirmed by a single-tube PCR method based on the amplification of internal transcribed spacer (ITS) regions as previously described by us (25).”

Reviewer 4 Report
The authors provide significant and relevant data about C. auris mortality. However, in order to improve their manuscript, the following issues should be addressed:
1) Scientific names appear throughout the text without italics. Please correct.
2) A paragraph describing the major problems with C. auris in a hospital setting (surface colonization and antifungal resistance) is welcome.
3) Why did the study finished in 2017? More recent data about this issue must be included. For instance, susceptibility profile of yeasts change , with emerging species appearing and spreading fast. The current profile of C. auris mortality may have changed, especially because the COVID-19 pandemic. In my oppinion, authors should gather data at least until 2021 to provide accurate, latest results on this subject.
4) Why did the authors choose to perform antifungal susceptibility using a commercial assay, instead of a standardized gold-standard method (CLSI, EUCAST)? Commercial assays often provide biased results, with associated errors. Moreover, why did the authors choose to interpretate their data using the CLSI M27-A3 breakpoints? Those breakpoints should be used with the CLSI M27-A3 method, another reason why the choice for the Sensititre Yeast One® AST plates was not the better one.
5) Why did the authors used the Student t-test to evaluate continous variables? Was normality of the data evaluated? These points should be addressed. Moreover, which confidence level was used in the analyses?
6) Please provide p-values with the decimal part separated by point, not comma, on Table 1. The same in line 132.
7) Columns total and range on Table 2 is not necessary. Put the total number of strains analyzed after the name of species (n=X).
8) The authors did not present an ethical statement for their paper. Was this study approved by an Ethical Review Board? Did the patients signed an informed consent? It is imperative to provide such information!
Author Response
- Scientific names appear throughout the text without italics. Please correct.
Answer: The scientific names were corrected
2) A paragraph describing the major problems with C. auris in a hospital setting (surface colonization and antifungal resistance) is welcome.
Answer:
The paragraph was added:
“C. auris has a high potential for outbreaks in healthcare settings, possibly due to environmental contamination or transient colonization by people or medical devices (3,4). This species is known to form biofilms on inert surfaces, which can persist for extended periods and require stringent disinfection processes to remove (5,6). To adapt to dry abiotic environments, C. auris activates stress-activated proteins (7) and produces hydrolytic enzymes that protect the yeast from environmental stressors and disinfectants like chlorhexidine and hydrogen peroxide. Chlorine-based products have shown to be the most effective for environmental surface disinfection (8–10).”
“C. auris is commonly resistant to azole drugs, and isolates that are resistant to all three main classes of antifungal agents have also been reported (19). The mutations in the drug-target lanosterol 14-α-demethylase ERG11 gene, such as Y132F, K143R, and F126L, frequently lead to azole resistance in C. auris strains. Additionally, mutations in the transcription factor TAC1 can cause overexpression of CDR1, which also leads to high-level azole resistance in C. auris (20).”
- Why did the study finished in 2017? More recent data about this issue must be included. For instance, susceptibility profile of yeasts change , with emerging species appearing and spreading fast. The current profile of auris mortality may have changed, especially because the COVID-19 pandemic. In my oppinion, authors should gather data at least until 2021 to provide accurate, latest results on this subject.
In a previous report, we described 20 cases of fungemia in hospitalized patients with severe acute respiratory syndrome coronavirus 2 (SARS-CoV-2) infection in four institutions in the northern region of Colombia from June to September 2020. Six patients had fungemia caused by C. auris, with four by C. albicans, four by C. tropicalis, three by C. parapsilosis, one by C. orthopsilosis, one by C. glabrata, and one by Trichosporon asahii. In this case series, the time to develop fungemia was similar to that reported in the current article (17.7 vs. 18.4 days). However, the mortality of this group of patients was higher (60% vs. 37.8%). The in vitro antifungal susceptibility of the six C. auris isolates was similar, showing high rates of resistance to FLC and AMB, whereas all were susceptible to echinocandins (unpublished data).
As mentioned earlier, the following paragraph has been added
“Although this study was conducted before the COVID-19 outbreak, we previously reported 20 cases of fungemia in hospitalized patients with severe acute respiratory syndrome coronavirus 2 (SARS-CoV-2) infection across four institutions in the northern region of Colombia between June and September 2020 (35). Of these, six patients had fungemia caused by C. auris. In this case series, the time to develop fungemia was similar to that reported in the current article (17.7 vs. 18.4 days). However, the mortality rate for this group of patients was higher (60% vs. 37.8%). Similarly, Villanueva-Lozano found a mortality rate of 83.3% (5/6) among patients with fungemia during an outbreak of C. auris infection in a COVID-19 hospital in Mexico (36), and Chowdhary found a mortality rate of 60% in patients with candidemia caused by C. auris who were critically ill with coronavirus disease and admitted to an intensive care unit between April and July 2020 in New Delhi, India (37). Therefore, it is expected that patients with fungemia caused by C. auris and SARS-CoV-2 infection may have a higher mortality rate.”
3 Why did the authors choose to perform antifungal susceptibility using a commercial assay, instead of a standardized gold-standard method (CLSI, EUCAST)? Commercial assays often provide biased results, with associated errors. Moreover, why did the authors choose to interpretate their data using the CLSI M27-A3 breakpoints? Those breakpoints should be used with the CLSI M27-A3 method, another reason why the choice for the Sensititre Yeast One® AST plates was not the better one.
Answer:
We chose to perform antifungal susceptibility testing using the Sensititre YeastOne® AST plates, which have been shown to be comparable to the CLSI reference method for testing the susceptibility of Candida spp (Pfaller MA, Chaturvedi V, Diekema DJ, Ghannoum MA, Holliday NM, Killian SB, Knapp CC, Messer SA, Miskou A, Ramani R. Comparison of the Sensititre YeastOne colorimetric antifungal panel with CLSI microdilution for antifungal susceptibility testing of the echinocandins against Candida spp., using new clinical breakpoints and epidemiological cutoff values. Diagn Microbiol Infect Dis. 2012 Aug;73(4):365-8. doi: 10.1016/j.diagmicrobio.2012.05.008).
The following paragraph has been added:
“Susceptibility to widely used antifungal drugs (AFG, anidulafungin; MCF, micafungin; CAS, caspofungin; 5-FC, flucytosine; PSC, posaconazole; VRC, voriconazole; ISV, isavuconazole; FLC, fluconazole; AMB, amphotericin B) was determined by the Sensititre Yeast One® AST plates (ThermoFisher Scientific, Les Ulis, France). This method was used because it remains comparable to the CLSI reference method for testing the susceptibility of Candida spp (26). For NACS isolates (C. albicans, C. tropicalis, C. glabrata and C. parapsilosis), the CLSI M27-A3 breakpoints were used (27). In the case of C. auris, breakpoints recommended by the US Centers for Disease Control and Prevention (CDC) were applied. Resistance to FLC was set at ≥32 µg/mL, AMB at ≥2.0 µg/mL and CAS at ≥2 µg/mL)(28).”
5) Why did the authors used the Student t-test to evaluate continous variables? Was normality of the data evaluated? These points should be addressed. Moreover, which confidence level was used in the analyses?
Answer:
According to Hoffman JE's recommendation, the student's t-test can be used to compare continuous variables of two groups. https://www.sciencedirect.com/science/article/pii/B978012817084700022X).
To evaluate normality, we use the statistical package GraphPad Prism 8.4.3.
According to the reviewer, we include the p values and confidence intervals for continuous variables to better clarify the results in table 1.
6) Please provide p-values with the decimal part separated by point, not comma, on Table 1. The same in line 132.
Answer:
As requested, the decimal separator for the p-value has been changed to point on table and text.
7) Columns total and range on Table 2 is not necessary. Put the total number of strains analyzed after the name of species (n=X).
Answer:
The columns "total" and "range" on Table 2 have been removed, and the number of strains analyzed has been placed after the species name.
8) The authors did not present an ethical statement for their paper. Was this study approved by an Ethical Review Board? Did the patients signed an informed consent? It is imperative to provide such information!
The paragraph was added:
“Ethical considerations: The research protocol was submitted and approved by the ethics committees of each participating institution. Since it was a retrospective study considered to be without risk, an exception was made for obtaining informed consent from all institutions. The analysis was conducted anonymously”

Round 2
Reviewer 4 Report
Authors claim in their response to have already published recent data about the mortality of C. auris infected patients. This already published result is different from the one they want to publish in the current manuscript. This does not make sense to me. Why do the authors want to publish that "mortality due to candidemia between C. auris and NACS was similar (abstract of the current manuscript)" if they already published that "the mortality of this group of patients was higher (60% vs. 37.8%, response letter)"?